# General Detection-based Text Line Recognition

**Raphael Baena, Syrine Kalleli, Mathieu Aubry**
LIGM, Ecole des Ponts, Univ Gustave Eiffel, CNRS
Marne-la-Vallée, France
`firstname.lastname@enpc.fr`

## Abstract

We introduce a general detection-based approach to text line recognition, be it printed (OCR) or handwritten (HTR), with Latin, Chinese, or ciphered characters. Detection-based approaches have until now been largely discarded for HTR because reading characters separately is often challenging, and character-level annotation is difficult and expensive. We overcome these challenges thanks to three main insights: (i) synthetic pre-training with sufficiently diverse data enables learning reasonable character localization for any script; (ii) modern transformer-based detectors can jointly detect a large number of instances, and, if trained with an adequate masking strategy, leverage consistency between the different detections; (iii) once a pre-trained detection model with approximate character localization is available, it is possible to fine-tune it with line-level annotation on real data, even with a different alphabet. Our approach, dubbed DTLR, builds on a completely different paradigm than state-of-the-art HTR methods, which rely on autoregressive decoding, predicting character values one by one, while we treat a complete line in parallel. Remarkably, we demonstrate good performance on a large range of scripts, usually tackled with specialized approaches. In particular, we improve state-of-the-art performances for Chinese script recognition on the CASIA v2 dataset, and for cipher recognition on the Borg and Copiale datasets. Our code and models are available at `https://github.com/raphael-baena/DTLR`.

## 1   Introduction

A popular approach in early Optical Character Recognition (OCR) was to localize individual characters before processing them independently [7]. Such an approach, while still used for specific cases such as Chinese script [56], cipher [41], or scene text recognition [58], has been largely discarded for Latin script Handwritten Text Recognition (HTR) in favor of implicit segmentation approaches. In this paper, we revisit character detection for text line recognition, demonstrate its effectiveness for HTR, and show results on a diverse range of datasets that we believe have not been demonstrated in prior work.

Designing a general detection-based approach for handwritten text recognition is challenging. Individual characters are often not well separated in handwritten texts, and they are not always readable independently of their context. Thus, most datasets provide only line-level annotations, without character bounding boxes. This comes in addition to the common challenges of text recognition – such as diversity in writers, very rare characters, different kinds of noise and degradation – and challenges unique to certain datasets, for which specific approaches have been designed, e.g., Chinese script which has a very large alphabet, or ciphers for which the amount of annotated data is limited. Most recent methods for HTR for Latin script rely on recurrent or autoregressive models, and very few are devoid of any recurrence [10]. However, we believe that a detection-based approach could be general, tackle HTR with any script, and provide potential advantages in terms of interpretability – since the position of each character is clearly identified, and errors can be explained by mis-localization or

38th Conference on Neural Information Processing Systems (NeurIPS 2024).

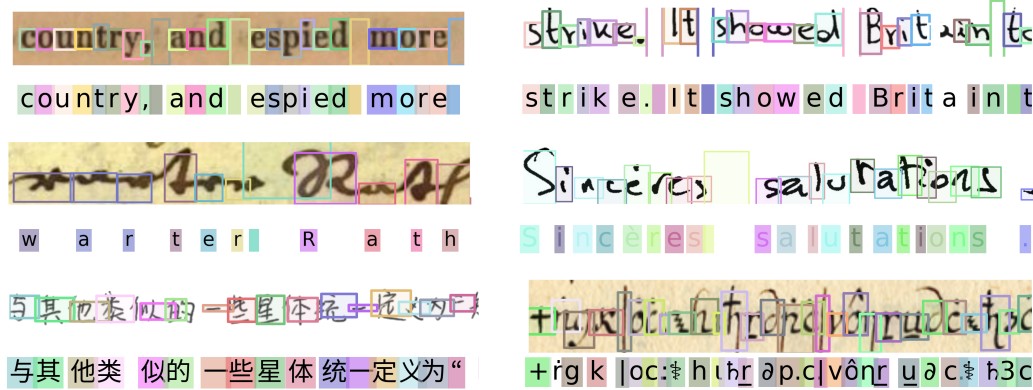

Figure 1: Our model is general and can be used on diverse datasets, including challenging handwritten script, Chinese script and ciphers. From left to right and top to bottom we show results on Google1000 [46], IAM [29], READ [44], RIMES [39], CASIA [27], Cipher [41] datasets.

mis-classification of ambiguous characters – as well as computational cost, since all characters can be processed in parallel.

We design such an approach, dubbed DTLR, by leveraging three critical insights. First, training with diverse and sufficiently challenging synthetic data enables the localization part of a detection network to generalize to characters unseen during training. Second, modern transformer-based detectors [6, 57] can detect all characters in a text line in parallel and allow for detections to interact with each other, especially if encouraged by an adapted masking strategy. Third, we introduce an approach to fine-tune a detection network end-to-end using only line-level annotations and demonstrate that it can be used even with a completely new alphabet.

Our contribution can be summarized as follows:

- we present the first transformer-based character detection approach for text line recognition.
- we demonstrate that our approach performs well on a wide range of datasets, including handwritten text in English, French, German, and Chinese text as well as ciphers.
- we significantly improve state-of-the-art performances for cipher recognition.

Our code and models are available at: `https://github.com/raphael-baena/DTLR`.

## 2 Related Work

Text recognition is the process of transcribing the text depicted in an image using computer algorithms. In this paper we focus solely on text line recognition, where a document image cropped around a line is given as input, which is the most standard setting in document analysis. In this section, we first give an overview of approaches for text line recognition, separating works focusing on Latin script, on Chinese script, and on ciphers, for which specific methods have been developed. We then discuss briefly the use of language models for text line recognition.

**Text line recognition for Latin script.** In the 1990s many methods leveraged character segmentation for OCR [18, 4] and later in the 2000s for HTR [16, 45, 2]. They involved several steps: (i) character segmentation, (ii) character feature extraction, and (iii) character classification [7]. The final predictions were refined, for example using hidden Markov models [16, 2] or grouping characters into words and matching them with a lexicon [18, 31]. Because explicit segmentation in HTR poses significant challenges [11], implicit segmentation techniques were developed, first using hidden Markov models [61, 8], then the Connectionist Temporal Classification (CTC) loss [21].

Leveraging the CTC loss [21] became the dominant approach in the field. Indeed, it provides a solution to one of the important challenge in text line recognition, the disparity in the lengths and the misalignment between the predicted output sequences and the ground truth sequences. Hybrid system combining Convolutional Neural Network and Recurrent Neural Network [40] were

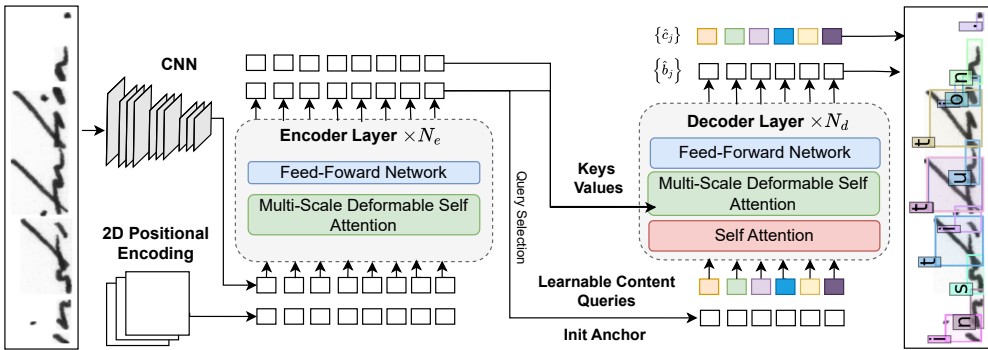

Figure 2: **Architecture.** Our architecture is based on DINO-DETR [57]. Given as input CNN image features, a transformer encoder predicts initial anchors and tokens, that are used by a transformer decoder to predict, for each token, a character bounding box and a probability for each character in the alphabet, including white space.

arguably the most successful, with many variations, for example using Long Short-Term Memory (LSTM) [20, 35, 5] or Multidimensional LSTM layers [47, 36]. Fewer studies used CTC with non-recurrent schemes [10, 55, 9], and are typically associated to lower performance [10].

More recently, still performing implicit character detection, the use of attention and transformers architecture enabled to complement [30] or replace [24, 13, 15, 14, 25, 19] the CTC loss with cross-entropy, using an autoregressive decoder: given the beginning of a transcription, the decoder predicts the next character using cross-attention with the image features. Wick et al. [51] extended this concept by fusing the outputs of a forward-reading transformer decoder with a backward reading decoder. Autoregressive decoders can handle text at various levels of tokenization, such as characters [13] or sub-words [25, 19]. Predicting one token at a time seems computationally sub-optimal, but these methods currently achieve the state-of-the-art performances [19].

In contrast, our approach performs explicit character detection and is neither recurrent nor autoregressive. It revisits an early OCR paradigm with modern transformer-based detection architectures.

**Text line recognition for Chinese script.** Explicit character segmentation, although no longer prevalent for Latin alphabets, remains an important approach for other languages, particularly Chinese HTR [53, 52, 33, 56]. Inspired by the YOLO framework [37], Peng et al. [33] proposed a detection network for character bounding box and class prediction in text line recognition. Building on this work, Peng et al. [34] introduced a weakly supervised method to train the detection network, using synthetic data with character-level annotations for pre-training, and generating pseudo labels for weakly labeled real data. Closer to our approach Yu et al. [56] trained a detection model on a synthetic dataset in a fully supervised way and used CTC loss for examples from their target dataset with only line level annotations, CASIA [27]. Note that most of these methods pretrain on a very large synthetic dataset of Chinese characters, while we use a general pre-training and learn Chinese characters only on a limited real training set with line-level annotations.

**Text line recognition for ciphers.** Recently, text recognition techniques have been used for transcribing historical ciphers, which can use different scripts, esoteric symbols, or diacritics. It is particularly challenging because for each cipher the number of documents and annotations is very limited, the underlying language is often unknown and no language model can be applied to the predictions [41]. Souibgui et al. [41] compared different approaches from HTR, including LSTM and transformer-based methods. Other approaches tried to explicitly segment or detect the characters. For example, Yin et al. [54] segmented the symbol and clustered them with a Gaussian Mixture Model, while Baró et al. [3] used K-Means. Antal and Marák [1] treated the problem as a character detection task but required character-level annotations for training.

**Language Model.** Models that incorporate recurrent schemes, such as Handwritten Text Recognition (HTR) with RNN layers [40, 36] or autoregressive decoders [13, 25], learn an implicit language bias that is crucial in HTR. Some models leverage very large amount of data to learn this language

prior. For instance, TrOCR [25] used 684 million text lines, and DtrOCR [19] two billion to train the decoders. Instead, we complement smaller scale training with a masking strategy, similar to, e.g., Lyu et al. [28]. The masking strategy most similar to ours is used in Chaudhary and Bali [10], where it is interpret solely as a form of data augmentation. Language models can also be used after the predictions have been made. Complex models can be trained for denoising, or with a masking strategy, e.g., Fang et al. [17], Wang et al. [49]. We used instead a simple N-gram [15, 42, 43] approach, which we train only on the training dataset. For common languages, better results could be expected by training more complex language models at a much larger scale, but this is out of the scope of our study, where we instead aim at generality.

## 3 Method

Given an input text-line image, our goal is to predict its transcription, i.e., a sequence of characters. We tackle this problem as a character detection task and build on the DINO-DETR architecture [57], shown in Figure 2, to simultaneously detect all characters. The rest of this section is organized as follows. First, in Section 3.1, we discuss how we pretrain a character-detection model using synthetic data with character-level supervision. Then, in Section 3.2, we explain how we finetune our model over real images from the target dataset with only line-level supervision and make final predictions.

### 3.1 Synthetic pre-training

**Data generation and masking strategy.** To generate our synthetic data, we start by defining an alphabet $\mathcal{A}_0$ and sampling sentences. We use two alphabets: one for Latin scripts and another for Chinese. The Latin alphabet contains 167 characters, including uppercase and lowercase Latin letters, whitespace, common symbols, and accented characters, while the Chinese alphabet includes 7,356 characters from the CASIA v1 dataset.

To simulate text line images, we build on the synthetic generation pipeline of Monnier and Aubry [32]. For the Latin alphabet, we sample a font from a set of publicly available fonts[1], using a font labeled as "handwriting" with a 50% probability, and a random background from a set of empty page photographs. We use the font to render the text and blend it with the background by using a random color and adding structured noise to both the background and font layers. For the Chinese alphabet, we use the CASIA v1 dataset [27], which contains various handwritten Chinese characters from different writers, to create synthetic sentences. We also apply significant blur data augmentation. This process results in challenging samples, as shown in the left column of Figure 3.

We sample sentences based on the application scenario. For known Latin-script languages, we sample sentences from Wikipedia in the target language. For ciphers, we generate random sentences by uniformly sampling Latin characters. For Chinese, we create random character sequences, since the limited alphabet of the CASIA v1 dataset makes it difficult to sample real Chinese text. This approach results in five models, for English, German, French, Chinese, and ciphers.

In addition, we use random erasing, masking complete vertical blocks, and small horizontal blocks completely, similar to Zhong et al. [59], Chaudhary and Bali [10]. This results in the text lines shown in the right column of Figure 3. Masking increases the robustness of the model, but also has additional motivations. First, masking vertical blocks can make a character unreadable or even completely hide it. In cases where the text is sampled from natural language, our network is thus encouraged to learn an implicit language model to predict detections for unreadable or even non-visible characters. Since our training time is modest, this model is far simpler than Large Language Models, but we still found it significantly improved predictions.

Second, masking small horizontal blocks will typically not make a character unreadable, but will prevent the model from focusing only on a discriminative part of a character. We found this to be especially important during the fine-tuning stage, where no character bounding box annotation is available.

**Architecture and loss** Since character labels and bounding boxes are known in synthetic data, we can simply leverage a standard DINO-DETR architecture [57], depicted in Figure 2 and the associated loss. Multi-scale image features are extracted through a CNN backbone and are then further refined

---

[1]https://github.com/google/fonts

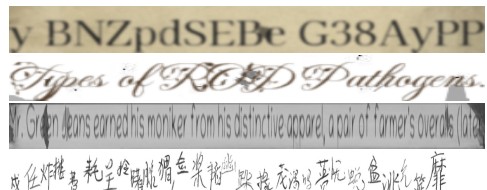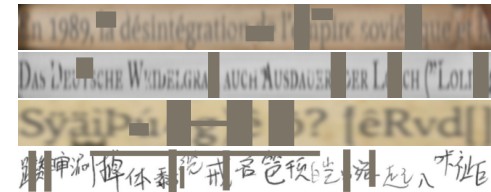

Figure 3: Samples from our synthetic datasets without (left) and with masking (right).

in the $N_e$ transformer encoder layers through deformable attention [60]. The transformer decoder is composed of $N_d$ decoder layers which are fed a set of $Q$ character queries. Each character query $q$ is the concatenation of a content query and a positional query, initialized by the output of the encoder. In each decoder layer, the character queries first interact through self-attention and then attend to the encoder image features via deformable cross-attention. The decoder outputs $Q$ bounding boxes $\hat{\mathbf{b}}_q \in \mathbb{R}^4$ and for each one a probability vector $\hat{\mathbf{c}}_q \in \mathbb{R}^{|\mathcal{A}_0|}$, where $|\mathcal{A}_0|$ is the number of characters in the alphabet $\mathcal{A}_0$ and $q$ is the query index.

For each image, the loss compares these $Q$ predictions to a set of $N$ ground truth bounding boxes $\mathbf{b}_n \in \mathbb{R}^4$ and their associated class label $\mathbf{c}_n \in \mathbb{R}^{|\mathcal{A}|}$, where $n$ is the index of the ground truth character. The box loss $\mathcal{L}_{\text{box}}(\mathbf{b}_n, \hat{\mathbf{b}}_q)$ measures the distance between a ground truth box $\mathbf{b}_n$ and a predicted box $\hat{\mathbf{b}}_q$ and is defined as a weighted sum between the $L^1$ distance and the generalized intersection over union (GIoU) : $\mathcal{L}_{\text{box}}(\mathbf{b}_n, \hat{\mathbf{b}}_q) = \lambda_1 \left\| \mathbf{b}_n - \hat{\mathbf{b}}_q \right\|_1 + \lambda_{\text{iou}} GIoU(\mathbf{b}_n, \hat{\mathbf{b}}_q)$. The number of queries $Q$ is typically much larger than the number of ground-truth characters $N$, so the set of ground truth is completed by $Q - N$ objects associated to the "no object" class $\emptyset$. The Hungarian algorithm is used to find a permutation $\hat{\sigma}$ between queries and ground truth minimizing a matching cost:

$$\hat{\sigma} = \arg \min_{\sigma} \sum_{i=1}^{Q} \lambda_{\text{cls}} \mathcal{L}_{\text{cls}}(\mathbf{c}_i, \hat{\mathbf{c}}_{\sigma(i)}) + \mathbf{1}_{\{c_i \neq \emptyset\}} \lambda_{\text{box}} \mathcal{L}_{\text{box}}(\mathbf{b}_i, \hat{\mathbf{b}}_{\sigma(i)}), \tag{1}$$

where $\mathcal{L}_{\text{cls}}$ is the focal loss [26] and $\lambda_{\text{cls}}$ and $\lambda_{\text{box}}$ are two scalar hyperparameters. Once this matching $\hat{\sigma}$ has been found, the network is trained to minimize the loss:

$$\sum_{i=1}^{Q} \lambda'_{\text{cls}} \mathcal{L}_{\text{cls}}(\mathbf{c}_i, \hat{\mathbf{c}}_{\hat{\sigma}(i)}) + \mathbf{1}_{\{c_i \neq \emptyset\}} \lambda'_{\text{box}} \mathcal{L}_{\text{box}}(\mathbf{b}_i, \hat{\mathbf{b}}_{\hat{\sigma}(i)}), \tag{2}$$

where $\lambda'_{\text{cls}}$ and $\lambda'_{\text{box}}$ are two scalar hyperparameters.

**Implementation details.** We follow Zhang et al. [57], and uses $N_e = 6$ encoder layers, $N_d = 6$ decoder layers, $Q = 900$ queries, and as hyperparameters $\lambda_{\text{cls}} = 2$, $\lambda_{\text{box}} = 5$, $\lambda'_{\text{cls}} = 1$ and $\lambda'_{\text{box}} = 5$. We generate synthetic datasets of 100k text lines, and train the networks for 225k iterations with batch size of 4, using the ADAM optimizer with $\beta_1 = 0.9$, $\beta_2 = 0.999$, a fixed learning rate of $10^{-4}$, and a weight decay of $10^{-4}$.

### 3.2 Finetuning with line-level annotations.

**Target alphabet.** The alphabet of the target dataset $\mathcal{A}$, might be different from the pretraining alphabet, $\mathcal{A}_0$. In our experiments, this is for example the case for ciphers. In this case, we replace the last linear layer responsible for character class prediction, adapting its size to the size of the target alphabet, and initialize each line, corresponding to a new character prediction, with random line of the pre-trained weight matrix, which we empirically found to work better than completely random initialization. Note the layers responsible for the bounding box coordinates predictions, which good initialization is critical for our approach, are unchanged. We actually found that thanks to the large variety of our training fonts, the character bounding boxes where reasonably good even for unseen characters. To train the new class prediction layer without destroying the rest of the network parameters, we fine-tune our network in two steps. First, for short a number of iterations (20k), we only optimize the re-initialized parameters, freezing the rest of the network. Then, we fine-tune the network end-to-end, thus also optimizing bounding box predictions.

Table 1: Character Error Rate (CER, in %) on standard HTR datasets.

| Method | IAM [29] | READ [44] | RIMES [39] |
|---|---|---|---|
| Sanchez et al. [38] | - | 5.10 | - |
| Michael et al. [30] | 4.87 | 4.70 | 3.04 |
| Diaz et al. [15] | 3.53 | - | 2.48 |
| Diaz et al. [15] with LM | 2.75 | - | **1.99** |
| PyLaia [2] | 8.44 | - | 4.57 |
| PyLaia+ N-gram [42] | 7.50 | - | 3.79 |
| VAN [12] | - | - | 3.04 |
| DAN [13] | 5.01 | **4.10** | 2.63 |
| DAN [13] + N-gram | 4.38 | - | 3.15 |
| TrOCR [25] | 2.89 | - | - |
| DTrOCR Fujitake [19] | **2.38** | - | - |
| Ours (DTLR) w/o. N-gram | 5.99 | 5.53 | 3.51 |
| Ours (DTLR) w. N-gram | 5.46 | 5.35 | 2.53 |

**From detection to sequences of character probabilities.** While most approaches to text recognition directly produce a sequence of character probabilities, our method outputs a set of detection without specific ordering and independent probabilities for all classes. For ordering the predictions, we simply sort them by the minimum $x$ coordinates of the predicted boxes. While this could cause issues with slanted scripts, we did not see any in practice. For simplicity, we assume in the rest of the paper that the queries $q$ are sorted with this order, which differs for each line.

The fact that our probability predictions are independent for each class, similar to DINO-DETR, makes it non-trivial to decide when to predict the "no object" class and makes it harder to compute CTC and combine our prediction with language models using standard tools such as Tarride and Kermorvant [42]. Given a predicted class probability $\hat{\mathbf{c}}_q \in \mathbb{R}^{|\mathcal{A}|}$, we thus define a new probability vector $\bar{\mathbf{c}}_q \in \mathbb{R}^{|\mathcal{A}|+1}$ by:

$$\bar{\mathbf{c}}_q^{|\mathcal{A}|+1} = \begin{cases} 1 - \sum_i \hat{\mathbf{c}}_q^i, & \text{if } \sum_i \hat{\mathbf{c}}_q^i < 1 - \varepsilon \\ \varepsilon, & \text{otherwise} \end{cases}$$

$$\text{and } \bar{\mathbf{c}}_q^i = \begin{cases} \frac{(1-\varepsilon)\hat{\mathbf{c}}_q^i}{\sum_i \hat{\mathbf{c}}_q^i}, & \text{if } \bar{\mathbf{c}}_q^{|\mathcal{A}|+1} = \varepsilon \\ \hat{\mathbf{c}}_q^i, & \text{otherwise} \end{cases},$$

where $\epsilon$ is a scalar hyper-parameter, we use upper indices to refer to vector coordinates and the sums are between 1 and the size of the alphabet $|\mathcal{A}|$. $\bar{\mathbf{c}}_q$ can now be interpreted as a joint probability vector for all classes, and $\bar{\mathbf{c}}_q^{|\mathcal{A}|+1}$ as the probability of the "no object" class.

**Fine-tuning with adapted CTC.** Unlike synthetic data, real-world datasets typically do not provide ground truth bounding boxes, but only line-level transcriptions. This makes it impossible to supervise character bounding boxes and to perform a Hungarian matching to associate tokens to characters. Thus, we instead adapt the CTC loss to fine-tune our network on real datasets. To do so, we leverage the sequence of character probabilities $\bar{c}$ defined in the previous paragraph. The standard CTC loss merges two successive identical characters, and requires them to be separated by a specific class, referred to as the *blank symbol*, to be separated. This is problematic for us since we might not predict empty bounding boxes between each character, and not necessary, since we detect each character independently and can directly use the bounding boxes to remove duplicate prediction using standard non-max suppression. We thus remove this behavior from the CTC loss. In practice, one can use standard implementations of the CTC loss and simply insert a blank symbol between each prediction.

Note that even with a novel alphabet, the network pre-training is critical, because if the predicted bounding boxes do not coarsely correspond to characters, training with the CTC is not enough for the network to learn meaningful character localization.

Table 2: Accurate Rate (AR) and Correct Rate (CR) [56] for Chinese HTR on CASIA [27].

| Method | Training Data | AR (%) ↑ | CR (%) ↑ |
|---|---|---|---|
| Wang et al. [48] | CASIA v1 + CASIA v2 | 90.20 | 90.80 |
| Wang et al. [48] | CASIA v1 + CASIA v2 | 90.23 | 90.80 |
| Peng et al. [33] | Synthetic (Chinese) + CASIA v2 | 90.52 | 89.61 |
| Wang et al. [50] | CASIA v1 + CASIA v2 | 92.18 | 90.77 |
| Yu et al. [56] | Synthetic (Chinese) + CASIA v2 | 96.77 | 96.77 |
| Ours (DTLR) | Synthetic (Chinese) + CASIA v2 | **96.83** | **97.34** |

Table 3: Symbol Error Rates (SER) and Word Accuracy (WA) for cipher recognition [41]

| Method | Copiale cipher | | Borg cipher | |
|---|---|---|---|---|
| | SER (%) ↓ | WA (%) ↑ | SER (%) ↓ | WA (%) ↑ |
| Text-DIAE [41] | 4.1 | 81.7 | 10.5 | 15.6 |
| Seq2Seq + Attention [41] | 3.6 | 82.8 | 10.7 | 54.7 |
| LSTM (VGG) [41] | 3.3 | 81.7 | 9.4 | 59.4 |
| LSTM [41] | 4.6 | 78.9 | 13.8 | 44.2 |
| Ours (DTLR) | **2.2** | **84.3** | **8.5** | **59.8** |

**Predictions refinement.** Once training is complete, the performance of the network can be slightly improved either by using non max suppression or by incorporating an explicit language model in the final decoding stage. We use the N-gram approach presented in [42] which combines N-gram probabilities with the CTC logits to include a prior on the likelihood of each text sequence. Similar to them, we find character-level models to be most suitable for our task as opposed to word-level models. However, unlike them, we found it beneficial to apply the N-gram word per word in each sentence, thereby making it similar to a spell-checking algorithm.

**Implementation details.** We use $\epsilon = 0.003$ to compute joint letter probabilities, and the same random erasing as for pre-training. We fine-tune our networks with the same parameters as for pre-training, except the learning rate for which we use $10^{-5}$ for 1200k iterations and then $10^{-6}$ for 800k iterations. For HTR in English, French and German, we train character-level $N$-grams on the text of the training set of each dataset with the KenLM [23] library and assign the N-gram a weight of 0.3 for each dataset. Otherwise, we use non-max suppression with an IoU threshold of 0.4.

## 4 Experiments

### 4.1 General text-line recognition

**Optical Character Recognition** We performed OCR on the Google1000 dataset [22], which contains scanned historical printed books, using our pre-trained model. We use the English Volume 0002, with 5,097 training lines, 567 validation lines, and 630 testing lines. Prior to fine-tuning on real data, our model has only $3.61\%$ in Character Error Rate (CER) and this rate is further reduced to $2.04\%$ after fine-tuning. Most of the "errors" are actually due to mislabeled samples. An example is shown in Figure 4. In Appendix, we also show qualitative results on the ICDAR 2024 Competition on Multi Font Group Recognition (Figure 8), a dataset that includes multiple fonts and languages, showcasing our method's ability to handle various languages and printing styles.

**Handwritten Text Recognition on Latin Alphabets** We evaluate our approach on text line HTR in various languages with latin script: IAM (English) [29], RIMES [39] (French), and READ [44] (Old German). For IAM, we follow common practice [13, 25, 19] and use the unofficial Aachen partition [3]. It includes 6,161 training lines, 966 validation lines, and 2,915 testing lines. The READ 2016 dataset consists of Early Modern German handwritten pages from the Ratsprotokolle collection.

---

[3] https://www.openslr.org/56/

It includes 8,367 training lines, 1,043 validation lines, and 1,140 test lines. The RIMES dataset is composed of administrative documents written in French. We use the RIMES-2011 version which includes only text lines from the letters' body [4]. The training set has 10,188 lines, which we split into a training set (80%) and a validation set (20%). The test set includes 778 lines. To assess the quality of our prediction, we use the Character Error Rate (CER) as defined by [13]. In Table 1, we present the results of our experiments across all three datasets.

On IAM, our results are satisfactory but clearly lower than state-of-the-art. The main reason seems to be that the dataset includes ambiguous samples which favors methods incorporating a more complex language model trained on large scale data [25, 19, 15]. Similarly, on the READ dataset, our performance is reasonable but still lower than the state-of-the art. This could be due to the challenges posed by the dataset: it has a small number of samples, a difficult handwriting style, and an old German language. In [13], the authors initially trained their model on synthetic lines generated from the READ dataset, which helps the model to learn the language prior. On the RIMES dataset, our performance is more competitive, and only outperformed by Diaz et al. [15] which relies on a large internal dataset for training. We observe that the simple addition of a character-level N-gram, trained solely on the training corpus, results in noticeable improvements for all datasets.

**Handwritten Text Recognition on Chinese**  To showcase the versatility of our approach, we also evaluate our method on CASIA v2 [27] a benchmark for handwritten Chinese text-line recognition, using Accurate Rate (AR) and Correct Rate (CR) as defined in [56]. The alphabet for this database has 2,703 characters, significantly more than Latin script. The training set consists of 41,781 text lines, which we further divide into a new training set (80%) and a validation set (20%) while the test set has 10,449 text lines. We report our results in Table 2. Our method outperforms the current state of the art by Yu et al. [56] in both accurate and correct rate.

**Ciphers**  We evaluate our approach on the Borg and Copiale cipher [5], using the line segmentation and splits outlined in [3]. The Borg cipher dataset [6], derived from a 17th-century manuscript, has 34 unique characters. The dataset is divided into a training set of 195 lines, a validation set of 31 lines, and a test set of 273 lines. The Copiale cipher, from 1730-1760, has 99 symbols and includes 711 training lines, 156 validation lines, and 908 test lines. As reported in Table 3, we achieve better results than the state of the art on both data sets. Figure 7 in the appendix presents challenging examples from the Copiale Cipher, where our method performs well.

DTLR also ranked first in the ICDAR 2024 Competition on Handwriting Recognition of Historical Ciphers [7] for the Borg and Ramanacoil ciphers. The competition featured five ciphers: a digit-based cipher, updated versions of the Borg and Copiale ciphers, enciphered documents from the Bibliothèque Nationale de France (BNF), and the Ramanacoil manuscript. The digit cipher uses 76 symbols, mostly digits, with various diacritics. The BNF cipher contains 37 unique graphical symbols. The Ramanacoil cipher uses 24 Latin-based symbols and special characters for seven key words. In Appendix, we report the competition results in Table 5, and show visual examples of our results on the Ramanacoil dataset in Figure 7.

## 4.2  Ablation and analysis

**Qualitative analysis**  Most of our results are visually meaningful, as can be seen in Figure 1. In Figure 4, we show the worst results for each dataset, which outlines the limitations of our model and the difficulty or problems in the datasets. On the Google1000 sample, the annotation is inaccurate since it is provided by an OCR model. On the IAM dataset, our model struggles to distinguish lowercase from uppercase letters in a specific portion of the test set where the test set letters are mostly uppercase, because the training set does not include such lines. On the RIMES sample, the input image is an example of a badly cropped line, a recurring problem in this dataset. The model misses some of the letters in `couvert` and relies on a misleading visual cue for é instead of ê. On both READ and CASIA v2, the model misses a few characters with high variation in writing style such as the `D` in READ or digits in CASIA v2 which are sparse in the training set of the model. On

---

[4]`https://zenodo.org/records/10805048`
[5]`https://pages.cvc.uab.es/abaro/datasets.html`
[6]`https://digi.vatlib.it/view/MSS_Borg.lat.898`
[7]`https://rrc.cvc.uab.es/?ch=27`

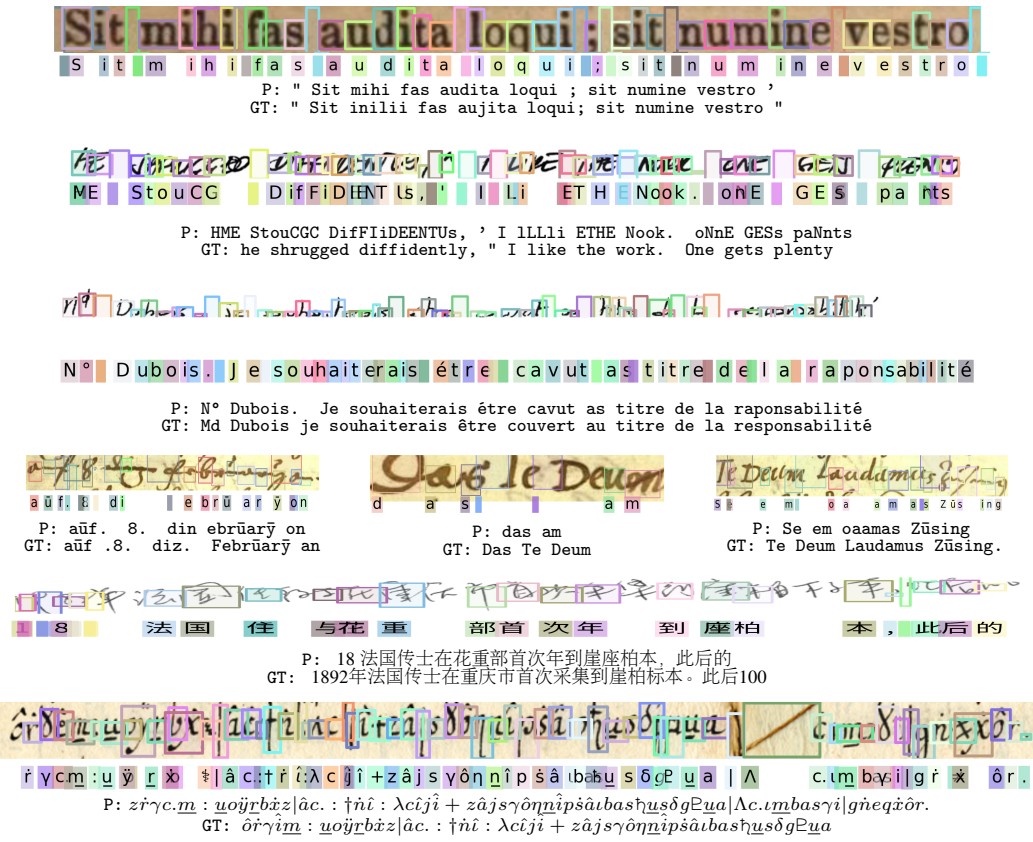

Figure 4: **Failure cases.** For the line(s) with the highest error on each dataset (Google1000 [46], IAM [29], RIMES [39], READ [44], CASIA [27], and Copiale [41]) we show, our detections, the predicted text (P) and the ground-truth text (GT). Best seen in color.

CASIA v2 where the fine-tuning process is much longer than for other datasets, we observed that the bounding boxes degenerate while the Character Error Rate (CER) continues to improve. This can be understood by the fact that no specific loss is used to constrain bounding boxes, and that features have large receptive fields. Thus, in Figure 1, we show the prediction of our model in middle of the training process, before the bounding boxes degenerate. On the Copiale dataset, the error comes from the labels, the last dozen characters are present in the image but not in the annotation.

**Ablation**    We conduct an ablation study in Table 4. We first examine the impact of the pretraining dataset on IAM, which uses latin script. Using a model trained on English and random erasing both improve performances. We then study the effects of the finetuning strategy as well as the presence of random erasing during finetuning on the IAM and Borg datasets.

Freezing the network and learning only the character classification layer significant boost performances on IAM, and results in a model that is better than random on Borg, but both still remain low quality, and end-to-end fine-tuning is necessary to obtain good performances. The results highlights the significant benefit of the random erasing strategy during finetuning, hinting that it helps our method to learn an implicit language model.

**Inference speed**    We compare our inference speed to TrOCr [25] and FasterDAN [14] for which public code is available, on text lines from the RIMES dataset [44] with a batch size of 1 and using an A6000 GPU. Our method requires 67ms for inference, while TrOCR takes 284ms and FasterDAN takes 140ms. However, FasterDAN is designed to recognize efficiently the lines of an entire document in parallel. When evaluating it on whole documents from RIMES and dividing the inference time by the average number of lines, one finds it requires only 7ms per line. While we can use a batch

Table 4: Ablation studies on the IAM and Borg dataset

| Method | IAM CER (%) ↓ | Borg SER (%) ↓ |
|---|---|---|
| General model | 39.72 | - |
| English model w/o. erasing | 37.20 | - |
| English model w. erasing | 36.70 | - |
| Finetuning only class embedding | 20.48 | 49.87 |
| w/o. erasing | 6.93 | 13.1 |
| w. erasing | 5.99 | 8.5 |

size 8, our inference speed decreases only to 25 ms, but one could imagine further improvements by adapting it to entire pages.

## 5 Conclusion

We presented a character detection approach to text line recognition. Although it is conceptually different from most modern HTR approaches, we demonstrated that it performs well on a wide diversity of benchmarks and achieves state-of-the-art performance for Chinese scripts and ciphers recognition. We hope that our general approach will revive detection-based approaches to text recognition and encourage the evaluation of future approaches on more diverse data.

## 6 Acknowledgments

This work was funded by ANR project EIDA ANR-22-CE38-0014, ANR project VHS ANR-21-CE38-0008, ANR project sharp ANR-23-PEIA-0008, in the context of the PEPR IA, and ERC project DISCOVER funded by the European Union's Horizon Europe Research and Innovation program under grant agreement No. 101076028. We thank Ségolène Albouy, Zeynep Sonat Baltaci, Ioannis Siglidis, Elliot Vincent and Malamatenia Vlachou for feedback and fruitful discussions.

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

# Appendix

Table 5: Symbol Error Rate (SER, in %) on ICDAR 2024 Competition on Handwriting Recognition of Historical Ciphers.

| Method | Digits | Borg | Copiale | BNF | Ramanacoil |
|---|---|---|---|---|---|
| Organizer - A | **7.83** | 7.91 | 4.33 | 1.09 | 6.07 |
| Organizer - B | 11.91 | 7.42 | 4.69 | 0.89 | 6.25 |
| S. Corbille | — | 7.60 | **1.62** | **0.89** | 6.08 |
| Jiaqianwen | 9.25 | 7.10 | 3.22 | — | — |
| Ours (DTLR) | 11.88 | **6.76** | 2.73 | 1.73 | **5.61** |

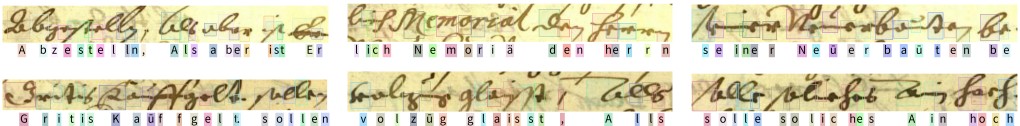

Figure 5: **Challenging Data.** Qualitative examples from the READ dataset with slightly slanted lines, degraded characters and translucent paper. Spaces are omitted for better visualization.

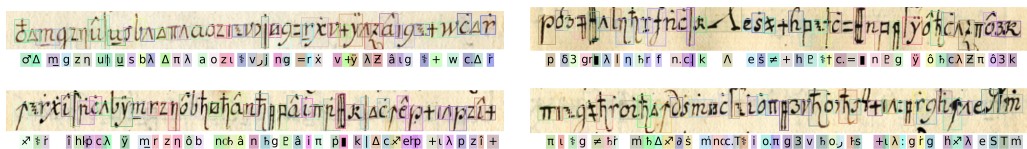

Figure 6: **Challenging Data.** Qualitative examples from the Copiale Cipher dataset with slightly slanted lines and translucent paper. Spaces are omitted for better visualization.

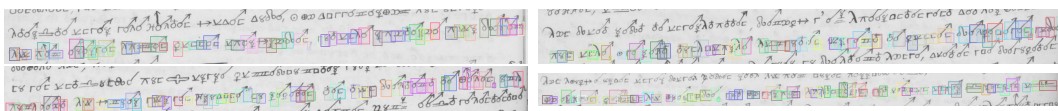

Figure 7: **Challenging Data.** Qualitative examples from the Cipher Ramanacoil dataset in the ICDAR2024 Competition on Handwriting Recognition of Historical Ciphers, where our model achieves perfect predictions.

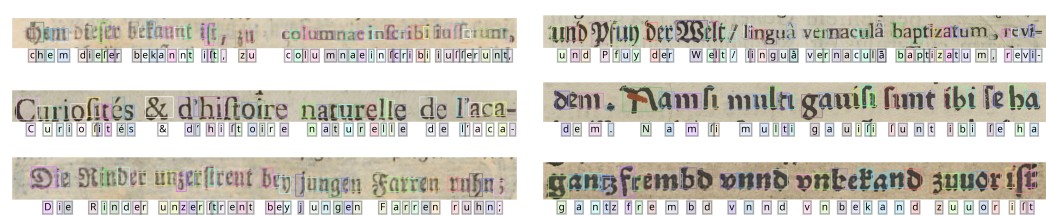

Figure 8: **Multilingual Predictions** Qualitative examples from the *ICDAR 2024 Competition on Multi Font Group Recognition and OCR dataset*, using a single model. The first row shows two examples of lines with mixed languages (German and Latin) and fonts. Subsequent rows display multiple languages (French, Latin, German) and fonts (Antiqua, Bastarda, Fraktur, Schwabacher) from left to right, top to bottom.

