# OpenReview forum: "General Detection-based Text Line Recognition"
_NeurIPS.cc/2024/Conference — NeurIPS 2024 poster_

### Official Review · Reviewer_jpUH · 2024-07-08

**Soundness:** 2
**Presentation:** 2
**Contribution:** 2
**Rating:** 6
**Confidence:** 4

**Summary:**

This manuscript provides a new approach to line recognition of offline handwriting and printed documents with regard to multilingualism.

This method can recognize a line based on its simultaneously recognized features based on transformers.

This approach is interesting for document analysis.

Promising results have been obtained for different datasets.

**Strengths:**

This paper is well written and organized. Since the method recognizes the characters simultaneously, this approach is efficient. This paper explores the impact of using a synthetic dataset to improve accuracy in practice.

**Weaknesses:**

This is not a whole system for the field of document analysis. This task is a part of document analysis that depends on line segmentation. This means that if the previous part (line segmentation) of the process has errors, these are transferred to the next steps. Therefore, this is an essential task for the work. What happens if you have a whole document for the task? What happens to your results if you use existing methods that segment lines?

This method is not generally suitable for multilingual contexts. If there are several characters in different languages, this paper has no clear solution. You should therefore have several models for each language.

It is better to explain the method for your synthetic data in more detail in the Appendix. It is better to clarify this with an illustration.

Please check your text again for some errors, e.g. line 263 (annotations annotations).

Is line 230 correct? "the English Volume 0002"

**Questions:**

When I compared your transformer architecture to the original [57], Figure 2 is different from the original. They use a query selection
 and matching. Why do not you have these parts? I can see them in your description, but not in your illustration.

You cite Figure 1 in subsection 4.2, why do you give the figure as Figure 1 without citing it earlier?

What happens if you have a document with multiple languages? Do you test your method for the type of input? Or should you train your network for this type of data?

Why don't you create the synthetic data for other languages and use a fine-tuning model based on English? Have you considered the scenario?

**Limitations:**

They showed some limitations in Figure 4. But it needs to add more samples in an Appendix.

---

> ### Author Rebuttal · Authors · 2024-08-06
>
> We thank the reviewer for their thorough review and insightful questions. We appreciate the comments on the organization and clarity of the paper, as well as the recognition of the efficiency of our method in recognizing characters simultaneously. We address the concerns and questions raised below.
>
> ## Weaknesses
> ### This is not a whole system for the field of document analysis. [...] What happens to your results if you use existing methods that segment lines?
>
> The most common practice in OCR and HTR is to evaluate text recognition on cropped text lines, not the entire pipeline. Very few papers address both tasks (e.g., DAN, FasterDAN).
>
> However, as detailed in the common rebuttal, we have begun to implement a model similar to ours, based on DINO-DETR, which is capable of detecting lines (Figure 1 of rebuttal PDF) with accuracy close to state-of-the-art in baseline detection, as shown in Table 1 of the rebuttal PDF. We use this model to detect lines on IAM and run our text recognition model. Performance is reported in Table 2.
>
>
>
> ### Please check your text again for some errors, e.g. line 263 (annotations annotations). Is line 230 correct? "the English Volume 0002"
> We thank the reviewer for noticing these errors. We have reviewed the text and made the necessary corrections.
>
> ## Questions
> ### When I compared your transformer architecture to the original [57], Figure 2 is different from the original. They use a query selection and matching. Why do not you have these parts? I can see them in your description, but not in your illustration.
>  Indeed, we use query selection at the end of the encoder as well as a denoising task for the decoder, similar to DINO-DETR. We initally removed these from Figure 2 for simplicity, but recognize that it can create confusion and will add them. We will correct the figure for the camera-ready version.
>
> ### You cite Figure 1 in subsection 4.2, why do you give the figure as Figure 1 without citing it earlier?
> Thank you for pointing this out. This Figure illustrates the capabilities of our method and the idea of our approach, we will refer to it explicitly in the introduction.
>
> ### What happens if you have a document with multiple languages?.
> Since our method does not heavily rely on language modelling (especially our general model), it is possible to use or fine-tune it on datasets that encompass multiple languages. We tested training a single model on a mix of Latin, French and Germanic printed manuscripts with 8 different historical fonts, and it works effectively with a cer of 1.68 % (note that better performances can likely be obtained by learning different models for different fonts, since different letters can have similar shapes in different fonts). We provide visual examples of results in Figure 5 of the rebuttal PDF,  and we will include more in the appendix.
>
>
> ## Limitation
> ### They showed some limitations in Figure 4. But it needs to add more samples in an Appendix.
>
> We will add an appendix to the paper with more examples of failure cases. We will also add more qualitative results similar to Figure 2, 3, 4 and 5 in the rebuttal pdf.

---

> > ### Comment · Reviewer_jpUH · 2024-08-13
> >
> > Thank you for your response. I will increase the rating.

---

### Official Review · Reviewer_KLkw · 2024-07-12

**Soundness:** 3
**Presentation:** 3
**Contribution:** 2
**Rating:** 6
**Confidence:** 4

**Summary:**

The paper introduces a novel transformer-based character detection for text line recognition. The authors use a diverse set of synthetic data to enable localization part of the detection network to generalize to unseen characters during training. The transformer-based detectors can identify all characters in a text line in parallel and a masking strategy has been adopted to encourage detection interactions. The methods proposed by the authors also includes a process to fine-tune the detection network using only line-level annotations.

**Strengths:**

S1: The approach is novel for character detection for text line recognition using transformer-based models.
S2: The authors demonstrate strong performance by the model across several datasets and outperformance for cipher recognition.
S3: The model can generalize well to unseen characters and variations in text as the authors utilize synthetic data.

**Weaknesses:**

W1: Several typos throughout the paper. Ex: typo in the Fig 1 caption: “Our model is general can be”,
W2: The authors can provide additional information on the fine-tuning process and the adaptability of the masking strategy. Did they explore other strategies?
W3: The performance on real-world datasets that contain some-to-large amount(s) of noise has not been extensively discussed by the authors.

**Questions:**

Q1: How does the model handle noisy datasets or texts with degraded quality?
Q2: What were the computational resource requirements? Can this approach be extended to real-world problems in a different setting?
Q3: How would LLMs affect or change the impact of this approach? Have the authors considered this aspect, and if the proposed method can do better than some existing LLMs?
Q4: Can this approach be extended to multi-line text detection and recognition?

**Limitations:**

L1: The efficiency of the proposed method in real-world datasets with noises has not been fully explored.

---

> ### Author Rebuttal · Authors · 2024-08-06
>
> We thank the reviewer for their thorough review and insightful questions. We appreciate their recognition of the originality of our method. Below, we address each of the reviewer's concerns and provide additional information to clarify our approach.
>
> ## Weakness
> ### W1: Several typos throughout the paper. Ex: typo in the Fig 1 caption: “Our model is general can be”,
> We thank the reviewer for pointing out these issues. We have carefully reviewed the paper and corrected the typos.
>
> ### W2: The authors can provide additional information on the fine-tuning process and the adaptability of the masking strategy. Did they explore other strategies?
>
> Regarding the masking strategy, we found empirically that combining both horizontal and vertical masking yielded the best results, similar to the approaches in [59, 10]. In addition, we used blur and random scaling as data augmentation strategies.
>
> Fine-tuning details, including the number of iterations and learning rate, are provided at the end of Section 3.2. Additional details on pretraining and finetuning will be included in an appendix, and the code will be released publicly.
>
> ### W3: The performance on real-world datasets that contain some-to-large amount(s) of noise has not been extensively discussed by the authors.
> We respectfully disagree with this assessment, as addressed in the common rebuttal. We have provided additional visual examples of READ and Copiale in Figures 2 and 3 of the rebuttal PDF, illustrating the challenges posed by these datasets.
>
> To further demonstrate the robustness of our model, we have included additional qualitative results for another cipher, Ramanacoil, in Figure 4 of the rebuttal PDF. Our model performs well on this dataset, that includes very slanted lines, leading to multiple lines being cropped jointly.
>
> We will include more examples for each dataset in the appendix. This appendix will also include visual examples from five other ciphers from the ICDAR2024 Competition on Handwriting Recognition of Historical Ciphers, which we did not include in the
> paper.
>
>
> # Questions
> ### Q1: How does the model handle noisy datasets or texts with degraded quality? Can this approach be extended to real-world problems in a different setting?
> As explained in response to W3, our method has been evaluated on several challenging real-world datasets, and we show additional results on challenging data in the rebuttal pdf. The method has been successfully applied to various settings, including printed text (google1000), handwritten text(IAM, READ, RIMES), multiple languages, and noisy data (ciphers, RIMES). It outperforms the state-of-the-art in cipher recognition and remains competitive on historical data (READ) that contains degraded quality text.
>
>
> ### Q2: What were the computational resource requirements?
> The pre-training took approximately one week, conducted on an RTX A6000 with 32 GB of memory, involving 225k iterations with a batch size of 4, as detailed in Section 3.1. The generation of synthetic data is expensive, accounting for 20% of the total time. Fine-tuning lasted 2 days, with details provided in Section 3.2. Iterations are faster during fine-tuning since no synthetic data is generated and the CTC loss is not costly to compute.
>
>
> ### Q3: How would LLMs affect or change the impact of this approach?
> 1. Multimodal LLMs capable of OCR cannot fully replace this approach. They will struggle with complex documents such as handwritten historical documents (e.g., READ) or ciphers that include rare symbols.
>
> 2. LLMs can complement our approach in several ways. For example, they can be used as external language models for post-processing to improve results, as demonstrated in [A]. However, this requires specific fine-tuning of the models. Indeed, we attempted to use ChatGPT 3.5 on the IAM dataset to refine predictions, but this approach was not successful due to the LLM hallucinating words, as the IAM dataset tends to split sentences and words. It is also possible to use an LLM to initialize part of the architecture, such as the decoder, as done in TrOCR. However, end-to-end training requires significant computational resources.
>
> [A] Thomas, Alan, Robert Gaizauskas, and Haiping Lu. "Leveraging LLMs for Post-OCR Correction of Historical Newspapers." Proceedings of the Third Workshop on Language Technologies for Historical and Ancient Languages (LT4HALA)@ LREC-COLING-2024. 2024.
>
> ### Q4: Can this approach be extended to multi-line text detection and recognition?
>
> We appreciate the reviewer's question regarding the extensibility of our approach to multi-line text detection and recognition. As detailed in the common rebuttal, we have begun to implement a model similar to ours, based on DINO-DETR, which is capable of detecting lines (Figure 1 of the Rebuttal PDF) with accuracy close to state-of-the-art in baseline detection, as shown in Table 1 of the rebuttal PDF. We use this model to detect lines on IAM and run our text recognition model. Performance is reported in Table 2.

---

> > ### Comment · Reviewer_KLkw · 2024-08-12
> >
> > Thank you for addressing my questions and comments. However, I believe the work requires some major revisions. Given this, I will maintain my score of 6: Weak Accept because the work presented is novel and interesting and the authors have given good explanation in the rebuttal. It just requires some major revisions.

---

### Official Review · Reviewer_9JMV · 2024-07-13

**Soundness:** 2
**Presentation:** 2
**Contribution:** 2
**Rating:** 6
**Confidence:** 5

**Summary:**

This paper presents a novel detection-based approach to text line recognition for both printed (OCR) and handwritten text (HTR), covering Latin, Chinese and cipher characters. Traditional detection-based methods have been largely neglected in HTR due to the difficulty of reading characters separately and the high cost of character-level annotation. The authors propose a solution to these challenges through three main insights: (i) using synthetic pre-training with diverse data for character localisation across different scripts; (ii) using modern transformer-based detectors to handle multiple character instances simultaneously, using a masking strategy to ensure consistency; and (iii) fine-tuning a pre-trained detection model with approximate character localisation using line-level annotations on real data, even with different alphabets.

**Strengths:**

- Originality: The method is highly original, proposing a detection/classification approach to character recognition that differs from most state-of-the-art methods, which typically rely on autoregressive decoding from images of lines or pages.
- Synthetic training data: The authors propose a way to train models using fully synthetic data, eliminating the need for actual character-level annotations.
- Model efficiency: The model is relatively small (~40M parameters), requiring only 100k synthetic line samples for training.
Error analysis: This approach allows for a better understanding of errors, distinguishing between detection and classification errors.
- Computational cost: Characters can be independently predicted in parallel, reducing computational cost, especially without a language model.
- Adaptability: The method can be easily adapted to any alphabet with minimal training data.
- Code release: The authors are committed to releasing the code, facilitating further research and application.

**Weaknesses:**

- Lack of new technical contributions: Despite the originality of the approach, the paper lacks novel technical contributions. The model, training strategy, fine-tuning with CTC, data augmentation and synthetic data generation are not novel.
- Performance on handwritten documents: The method is not competitive on Latin handwritten documents and is outperformed by existing methods on Chinese handwritten documents. It does outperform on cipher recognition, but this is a more specialised and less researched area.
- Scope of evaluation: The framework is only evaluated on perfectly segmented lines of text, raising questions about its performance on full pages. Full page processing would require additional steps such as text line detection and reading order retrieval, which could impact performance.
- Impact of line detection: There are concerns about how the quality of line detection would affect character detection and recognition, particularly in cases where vertical or horizontal lines merge.

**Questions:**

- Error cases: Can you provide examples of challenging real-world examples, such as rotated, upside-down or slanted lines; blank lines; vertically merged lines; strikethrough text; translucent paper; and mixed printed/handwritten characters?
- Pre-training: How long was the model trained, on what hardware (GPUs), and why was pre-training limited to 100k lines of text? Was this value determined experimentally?
- Input image size: What was the input image size used for training and evaluation?
- Computational cost: How do you explain the difference in computational cost between DINO-DETR (25ms/line) and FasterDAN (7ms/line)? Why were comparisons made using a batch size of 1? Please provide a more comprehensive comparison, including batch sizes, CPU vs. GPU performance, impact of language modelling, input image sizes, and model parameters.
- Language model decoding: What is the impact of a language model on inference speed? Can character recognition/classification still be parallelized with a language model? Can you decode on GPU with a KenLM language model, and what size of N-gram language model do you use?
- Model release: Will the model be released to the public?

**Limitations:**

- Conclusion: The conclusion lacks depth and provides no insight into future improvements to the method.
- Paper and writing:
    - Figure 2: The figure is unclear and needs to be made clearer.
    - Table 1: The table is misplaced, it appears on page 6 but is referenced on page 7.

---

> ### Author Rebuttal · Authors · 2024-08-06
>
> We thank the reviewer for their thorough review. We are glad they found the paper highly original, efficient, and adaptable. We address the concerns and questions raised below.
>
> ## Weakness
> ### Lack of new technical contributions.
> We respectfully disagree with this assessment, as detailed in the common rebuttal and the answer to R-hupU.
>
> ### Performance on handwritten documents.
> We respectfully disagree as answered in the common rebuttal.
>
>
> ### Impact of line detection.
> We agree that the method used to extract text lines has an impact on performance. However, evaluating on cropped text lines is the most common practice in OCR and HTR, rather than evaluating an entire pipeline. Very few papers  (e.g., DAN, FasterDAN) address both tasks.
>
> In cases where the line detection quality is poor and multiple lines appear in one crop, our method can still learn to detect only the characters of the line of interest, as shown in Figure 4 of the rebuttal PDF.
>
> ## Question
> ### Error cases.
>
> In the paper, our method has been evaluated on several challenging real-world datasets with such issues. In Table 1 of the paper, we report results on the READ and Copiale datasets, which include translucent paper and slanted lines, as shown in Figures 2 and 3 of the rebuttal PDF. IAM and RIMES datasets include strikethrough text, which we will include in an appendix.
>
> Our method should perform well on mixed printed and handwritten text, as it has been effective on both printed (Google 1000) and handwritten datasets (IAM, READ, RIMES, HWDB), and on datasets mixing multiple fonts and languages, as shown in Figure 5 of the rebuttal PDF.
>
> We see "rotated, upside-down" text as out of the scope of the paper. We consider that correctly detecting the orientation and shape of such lines is the role of the line detector. Lines can be slightly slanted, as is the case in several of our datasets and the one reported in Figure 4 of the rebuttal pdf.
>
> ### Pre-training.
>
> The pre-training took approximately one week, conducted on an RTX A6000 with 32 GB of memory, involving 225k iterations with a batch size of 4, as detailed in Section 3.1. The generation of synthetic data is expensive, accounting for 20% of the total time. Finetuning lasted for two days (number of iterations are given in section 3.2).
>
> The number of synthetic lines was not tuned, we simply generated a high number of lines. We believe that using a very large number of text lines during pre-training is not necessary since our method only learns a limited implicit language model. However, we will explore the impact of the synthetic training dataset size systematically in the final version (since this requires re-training several models, this was not doable during the time of the rebuttal).
>
>
> ### Computational cost.
>
> As explained in Section 4.2 we evaluated the inference speed of FasterDAN on whole documents, and the inference time was divided by the average number of lines. It is not a fair comparison for our method since our model does not work on whole documents while FasterDAN does. As explained in the common rebuttal, we aim to adapt our model for whole documents by using an additional model to guide attention, thereby reducing inference time like Faster DAN.
>
> When we launch both methods on individual lines, our method requires 67ms for inference, while FasterDAN takes 140ms.
>
> **Batch Size:**
> Comparisons were made using a batch size of 1 to ensure a fair evaluation of inference time. Below, we provide a table comparing inference speeds in ms across different batch sizes:
>
> | Batch Size | 1 | 2 | 4 |  8 | 16 | 32 |
> |-|-|-|-|-|-|-|
> | Ours | 74 | 44 |34 |31 |28 |32 |
> | TrOCR | 271 | 208 | 111 |59 |49 |46 |
> | FasterDAN Document / average number of lines| 56 | 35 |21 |16 |16 |21 |
> | FasterDAN Lines | 140 | 74 |28 |16 |8 |7 |
>
> **Image Sizes:**
> We agree that image size impacts speed. However, text lines are approximately the same size and are resized to a maximum width of 1330 pixels.
>
> **Model Parameters:**
> We agree that model parameters impact both inference time and performance. But, analyzing these effects in detail would require extensive experimentation. Our focus was on demonstrating the effectiveness of our method in handling text recognition and character segmentation efficiently.
>
> ### Language model decoding.
>
> We employ the KenLM library and pytorch CTC decoder which do not support GPU acceleration. We have measured the inference speed on the RIMES dataset on a single CPU to be 960 ms per example. While it could be parallelized using CPU multiprocessing, this optimization is beyond the scope of our current work. Our aim was to illustrate that our predictions can be further refined through the use of a language model. One could perfectly employ a LLM that supports GPU decoding.
>
> The N-gram model used in our experiments is of size 6, tuned on the validation set.
> ### Model release: Will the model be released to the public?
>
> Yes, as stated in the introduction, all models will be released publicly, including pre-trained and fine-tuned models.
>
> ## Limitations
> ###  The conclusion lacks depth and provides no insight into future improvements to the method.
> We can extend the conclusion by adding the following future directions:
>
> 1. **Text Recognition on whole documents**:  detailed in the common rebuttal.
> 2. **Advanced Language Models**: Improving recognition by integrating more advanced language models, such as a GPT decoder.
> 3. **Unsupervised Learning**: Implementing character segmentation with a reconstruction loss, allowing the model to learn the characters (alphabet) of a dataset and perform text recognition without any annotations, similary to [A].
>
> [A] Siglidis, Ioannis, et al. "The Learnable Typewriter: A Generative Approach to Text Analysis." arXiv preprint arXiv:2302.01660 (2023).
> ### Paper and writing
> As addressed in our response to RjpUH, we will clarify Figure 2 and have corrected the placement of Table 1.

---

> > ### Comment · Reviewer_9JMV · 2024-08-13
> >
> > Thank you for the detailed answer. I'm convinced of the value of the proposed method, mainly because of its originality compared with current methods. Without being revolutionary, it opens up interesting perspectives and I would like to see it tested further. The availability of the code and models will allow this. I'm going to raise my score.

---

### Official Review · Reviewer_hupU · 2024-07-17

**Soundness:** 3
**Presentation:** 3
**Contribution:** 2
**Rating:** 6
**Confidence:** 4

**Summary:**

This paper treats text line recognition as an object detection task and proposes a two-stage training approach based on DINO-DETR. In the first stage, synthetic data with bounding box information is used to predict the bounding boxes and categories of text, due to the absence of character-level annotations for text line datasets. In the second stage, real text line data is employed to sequentially fine-tune the classifier and the entire model. The proposed method achieves state-of-the-art performance in cipher text.

**Strengths:**

- This paper is well-written and easy to follow.
- The evaluation is well done and compares against several strong baselines.

**Weaknesses:**

- Abalation study is easy.
- Utilizes more engineering skills than methodological innovations.

**Questions:**

All results in the paper were pre-trained on Latin text and fine-tuned on several datasets. What if we pre-trained and fine-tuned on data in the same language? For example, pre-training on synthetic Chinese text lines followed by fine-tuning on real Chinese datasets.

**Limitations:**

The proposed method performs well only on the cipher text recognition task.

---

> ### Author Rebuttal · Authors · 2024-08-06
>
> We thank the reviewer for the detailed review and positive feedback on the paper, in particular positive comments on the clarity of the paper and the thorough evaluation against strong baselines. We address the concerns and questions raised below.
>
> ## Weakness
> ### Abalation study is easy.
>
> We evaluate in our ablation the impact of the pre-training dataset, the impact of erasing during pre-training and fine-tuning, and we demonstrate the importance of fine-tuning the complete network, including the bounding box prediction, despite the fact there is no supervision on bounding boxes during fine-tuning. We will happily add any other ablation that the reviewer thinks would clarify aspects of our method.
>
> ### Utilizes more engineering skills than methodological innovations.
> We respectfully disagree with this assessment. As explained in the common rebuttal, building an OCR/HTR method based on a transformer detection architecture requiered methodological innovations and not mere engineering, which we believe is the reason why it was never demonstrated. While the pre-training losses are the same as DINO, the fine-tuning with CTC for such an architecture is novel and requires (i) ordering the queries differently for each text line according to the bounding boxes, (ii) adapting the class probabilities produced by the DINO architecture (iii) adapting the CTC loss, which can be done by introducing additionnal blank tokens, as detailed in our methodology section.
>
> We believe that it is due to these methodological contributions, and not heavy engineering, that our method is, to the best of our knowledge, the first method competitive on many datasets, performing jointly text recognition and character segmentation, and processing characters in parallel rather than in an autoregressive way.
>
> ## Questions
> ### All results in the paper were pre-trained on Latin text and fine-tuned on several datasets. What if we pre-trained and fine-tuned on data in the same language? For example, pre-training on synthetic Chinese text lines followed by fine-tuning on real Chinese datasets.
>
> We used pre-trained models for each language (French, German, English) and one general model trained on random text for the cipher and Chinese datasets. We noticed that this information was missing from Section 3.1. We will include all the information in the camera-ready version.
> As shown in Table 4 of the ablation study, pretraining the model and the target language results with a masking strategy results to better results on IAM. We will compare performance with generic and language specific pre-training for all datasets in the final version of the paper.
>
> Regarding Chinese, we have pre-trained a model on Chinese characters and are currently fine-tuning it on CASIA. We achieved an AR of 94.3 and a CR of 95.3, surpassing the performance reported in the paper by  2.1\%.
>
>
> ## Limitations
> ### The proposed method performs well only on the cipher text recognition task.
>  We refer the reviewer to the common rebuttal where we address this limitation.

---

### Author Rebuttal · Authors · 2024-08-06

We thank the reviewers for their thoughtful feedback. We are pleased that our approach was found to be highly original (R9JMV), novel (RKLkw), efficient (R9JMV, RjpUH), rigorously evaluated (RhupU, RKLkw), and well written (RhupU, RjpUH). We address in the following main concerns, raised by several reviewers.

## Lack of Technical Novelty (RhupU, RKLkw)

We respectfully disagree with this assessment. Building an OCR/HTR method based on a transformer detection architecture, allowing for parallel processing of characters instead of the sequential processing typical in current methods [13, 15, 19, 25, 30, 38], requiered methodological innovations and not mere engineering, which we believe is the reason why it was never demonstrated. While the pre-training losses are the same as DINO-DETR, the fine-tuning with the CTC loss for such an architecture is novel and requires several ideas and contributions: (i) ordering the queries differently for each text line according to the bounding boxes, (ii) adapting the class probabilities produced by the DINO-DETR architecture, (iii) adapting the CTC loss, which can be done by introducing additionnal blank tokens, as detailed in our methodology section.

We believe that it is due to these methodological contributions, and not heavy engineering, that our method is, to the best of our knowledge, the first method competitive on many datasets, performing jointly text recognition and character segmentation, and processing characters in parallel rather than in an autoregressive way.

##  Lack of Evaluation on Challenging Real Datasets (R9JMV, RKLkw)

Our method has been evaluated on the historical READ dataset, which includes Germanic manuscripts and two ciphers. These datasets are characterized by noise, degradation, and can have slightly slanted lines as shown in Figure 2 (READ) and 3 (Copiale) of rebutal PDF. We have also included additional examples from two extra challenging datasets: the Ramancoil cipher (Figure 4) and a collection of early-modern prints from the
ICDAR24 Competition on Multi Font Group Recognition and OCR (Figure 5).

The Ramancoils cipher (year 1674) includes very slanted lines and the crops can include multiple lines, as shown in Figure 4 of the rebuttal PDF. Our model successfully recognizes the correct line even despite the presence of the upper and lower lines.

The early-modern prints dataset features eight different fonts and three languages (Old French, Old German, and Latin). We trained a single model capable of handling these diverse fonts and languages, as illustrated in Figure 2. The model also processes lines containing mixed fonts and languages (German and Latin) within the same line, as shown in Figure 5. Note that better performances can likely be obtained by learning different models for different fonts, since different letters can have similar shapes in different fonts.

## Competive results limited to Ciphers

We respectfully disagree. Our approach demonstrates strong performance across the datasets mentioned in the paper. While our results on the IAM dataset are below the state-of-the-art, it is important to note that leading methods like DTRoCR and TrOCR use significantly larger architectures than ours with up to 10 times more parameters, extensive computational resources (e.g., TrOCR used 32 V100 GPUs and a batch size of 1024), and much larger-scale training datasets (hundreds of millions of printed text lines for TrOCR). We believe that given comparable computational resources, similar performances could be achieved by our model.

Our method also provides a valuable contribution by efficiently addressing both text recognition and character segmentation, which is not achieved by any current method.

Following the suggestion of RhupU, we pre-trained a new model on synthetic lines of Chinese using characters from CASIA v2. We are currently fine-tuning this model on CASIA v2 and have achieved an AR of 94.3\% and a CR of 95.3\%, which is already  2.1\% above the performance reported in the paper, reducing the gap with state-of-the-art. Moreover, the training loss has not had time to converge during the rebuttal period, and better results can be expected after convergence.

##  Text Recognition on whole documents

We acknowledge the value of a system that can handle both text detection and recognition. However, the standard in text recognition typically involves evaluating crops of individual lines [19, 25, 30, 38]. Few methods address this combined task [12, 13].

Nevertheless, we provide proof of concept results of a model similar to ours, derived from DINO which performs text line detection in the rebuttal PDF. This method predicts 8 points for the baseline of the text and 2 for the line above it to segment the line, enabling both baseline evaluation on standard datasets and bounding-box extraction. It is pre-trained on synthetic data and fine-tuned on real data. The method achieves results close to the state of the art in baseline detection, as reported in Table 1 of the Rebuttal PDF. We also provide visual examples of line predictions on complex datasets (cBAD2019) and IAM in Figure 1 of the rebuttal PDF. Finally, we provide CER for combined line detection and recognition on IAM in Table 2.

This is a promising first step, but it could be improved by fully integrating it with our approach and training it end-to-end by using the line detections to guide the attention weights of our text recognition model. Recognition for several lines could then be performed in parallel, similar to FasterDAN. Similar to our model, pretraining could be performed in a completely supervised way, while fine-tuning could be done with only page level supervision. However, we believe that demonstrating a detection-based model for text lines is a significant enough contribution for this paper, and introducing a complete pipeline would dilute the contribution and limit the diversity of datasets on which the full approach can be evaluated.

---

### Decision · Program_Chairs · 2024-09-25

**Decision:**

Accept (poster)

**Comment:**

The paper introduces a novel transformer-based handwritten character detection for text line recognition. The authors use a diverse set of synthetic data to enable localization part of the detection network to generalize to unseen characters during training. The transformer-based detectors can identify all characters in a text line in parallel and a masking strategy has been adopted to encourage interactions  between detections. The methods proposed by the authors also includes a process to fine-tune the detection network using only line-level annotations.

All reviewers support weak accept (6), some of them having increased their scores during the rebuttal.

Most reviewers agree that this paper is well written and supported by solid evaluations that compare well to the state-of-the art. While it revisits an earlier OCR paradigm, this approach is very original when compared to current mode holistic handwritten recognition approaches based on auto-regressive models. Compared to those, it seems much more efficient in terms of inference cost and also training cost. Synthetic data could make training on low resource languages more accessible.

The main weakness mentioned by 2 reviewers (9jmv, hupu) is the limited technical contribution. The author solid rebuttal, showing that they use the computer vision DINO-DETR model in a novel way in combination with CTC, caused one of the reviewers to increase his score.
Another limitation is the absence of real world applications, going from single lines to full documents. This would have allowed the authors to show how more cost effective their approach is.

This paper deserves a weak accept due to the limited technical contribution, but it could have a significant impact on the field given the originality and efficiency of the method